# Shortcut Solutions Learned by Transformers Impair Continual Compositional Reasoning

## Abstract

Identifying and exploiting common features across domains is at the heart of the human ability to make analogies, and is believed to be crucial for the ability to continually learn. To do this successfully, general and flexible computational strategies must be developed. While the extent to which Transformer neural network models can perform compositional reasoning has been the subject of intensive recent investigation, little work has been done to systematically understand how well these models can leverage their representations to learn new, related experiences. To address this gap, we expand the previously developed Learning Equality and Group Operations (LEGO) framework to a continual learning (CL) setting ("continual LEGO"). Using this continual LEGO experimental paradigm, we study the capability of feedforward and recurrent Transformer models to perform CL. We find that BERT, a canonical feedforward Transformer model, learns shortcut solutions that limits its ability to generalize and prevents strong forward transfer to new experiences. In contrast, we find evidence supporting the hypothesis that ALBERT, a recurrent version of BERT, learns a `For` loop-esque solution, which leads to better CL performance. When applying BERT and ALBERT models to a CL setting that requires composition across experiences, we find that both model families fail. Our investigation suggests that ALBERT models can have their performance drop rescued by use of training strategies that combine data across experiences, but this is not true for BERT models, where a detrimental shortcut solution becomes entrenched with initial training. Our results demonstrate that the recurrent ALBERT model may have an inductive bias better suited for CL and motivate future investigation of the interplay between Transformer architecture and computational solutions that emerge in modern models and tasks.

## 1 Introduction

At the core of compositional reasoning (i.e., solving complex tasks by iteratively solving and composing simpler sub-tasks) is the ability to recognize relationships between objects. While the exact objects and their relationships may differ across different types of experiences, common underlying structure can be broadly present. Therefore, developing general computational strategies for composition not only enables good performance on individual experiences, but can also allow for faster acquisition of new experiences. For instance, learning to iteratively loop over objects and their relationships (i.e., an algorithmic `For` loop-esque computation) can be equally useful when spatially navigating (e.g., "turn left at the stop sign", "turn right at the light",...) and when solving a logic problem (e.g., "Alice has four watermelons", "Alice gives two watermelons to Bob",...). Thus, endowing artificial neural networks with the ability to perform compositional reasoning over continually new experiences will involve identifying how to endow them with general computational strategies.

Self-attention (Bahdanau et al., 2014) has revolutionized machine learning by enabling the learning of input-specific relations across long context windows. Analysis of the emergent attention patterns has identified interpretable and generalizable computations (Zhang et al., 2022; Kantamneni et al., 2024). For instance, dissection of the attention patterns learned by BERT (Devlin et al., 2018), a canonical feedforward Transformer model, and ALBERT (Lan et al., 2019), its recurrent counterpart (with weights shared between

layers), on the synthetic Learning Equality and Group Operations (LEGO) task unveiled the presence of attention heads that performed local and global attention (Zhang et al., 2022). Mimicking these patterns in randomly initialized networks leads to increased performance, emphasizing the learned computational strategies' utility and suggesting possible routes to improving upon standard Transformer architectures.

While demonstrating the power of Transformers, self-attention has also been found to support "shortcuts" (Liu et al., 2023) (i.e., solutions that are non-generalizable). Indeed, it has been proven that such shortcut solutions always exist in the context of certain tasks (Liu et al., 2023), suggesting they may be inescapable. However, there is also evidence that Transformer models can learn generalizable solutions, as ALBERT was argued to learn a `For` loop computation (Zhang et al., 2022). This tension between general and shortcut solutions has major implications for if and how Transformer models are capable of learning to continually compose across new experiences. And yet the fundamental capabilities of Transformer models to perform continual learning (CL) remains a largely understudied research direction, particularly in the context of compositional reasoning (although see Abdool et al. (2023) for some recent work in this direction).

To begin to address this important gap in a simplified and controlled setting, we develop a CL extension of LEGO ("continual LEGO") and perform an in-depth analysis on how BERT and ALBERT models perform. Our work provides the first in-depth analysis of how feedforward and recurrent Transformer models perform on continual compositional reasoning. Our contributions are the following:

- We expand the synthetic compositional reasoning LEGO task (Zhang et al., 2022) to enable a systematic investigation of CL capabilities of Transformer models.

- We find that architectural choices (e.g., number of attention heads, number of hidden layers) differentially affect the generalization accuracy and strength of forward transfer for BERT and ALBERT models, with BERT models demonstrating inconsistent performance on CL, as model size is increased.

- We identify evidence of shortcut and algorithmic `For` loop-esque solutions in BERT and ALBERT models, respectively, providing a mechanistic explanation of their different performance on the continual LEGO task.

- We demonstrate that both families of models fail to perform well on a continual LEGO setting that requires composition across experiences, a failure that we find can be rescued by training on data that incrementally combines across experiences in ALBERT models, but not BERT models, where a shortcut solution has become entrenched.

Collectively, our results demonstrate that – on a synthetic continual compositional reasoning task – the shortcut solutions learned by BERT and ALBERT models, while enabling success on complex tasks, can impair their CL capabilities. We hope our work leads to greater understanding of these important limitations and to what extent they exist in modern Transformer architectures and tasks.

## 2 Related work

### 2.1 Continual learning with Transformers

CL has historically been studied in the context of computer vision (Van de Ven et al., 2022). Unsurprisingly then, vision Transformers (ViTs) (Dosovitskiy et al., 2020; d'Ascoli et al., 2021) have received the bulk of attention from investigations characterizing Transformers' ability to perform CL (Yu et al., 2021; Wang et al., 2022; Zheng et al., 2023). This work has demonstrated that ViTs are susceptible to catastrophic forgetting, in some cases even more so than CNNs (Yu et al., 2021). This failure has been attributed to degradation of the locality that emerges in the self-attention heads (Zheng et al., 2023), as well as a greater bias towards new classes (Yu et al., 2021). Addressing these limitations by architectural changes leads to improvement in CL performance (Yu et al., 2021; Wang et al., 2022; Zheng et al., 2023). For transformer-based large language models Yıldız et al. (2024), recent work has benchmarked CL when sequentially training on language corpuses across domains for different model sizes. Our work complements this prior literature by considering

CL in the context of compositional reasoning, which to our knowledge has received little investigation from the Transformer community (Abdool et al., 2023). Additionally, unlike image classification, reinforcement learning, or language domains where distributional shifts can be challenging to quantify, our continual compositional reasoning tasks offer a structured approach to study forward transfer. Thus, we are able to probe forward transfer and emergent computational solutions to a greater extent than what is typically studied in CL.

## 2.2 Compositional reasoning with Transformers

A growing body of work has begun to investigate the mechanisms by which Transformers perform compositional reasoning (Geiger et al., 2021; Zhang et al., 2022; Allen-Zhu & Li, 2023; Li & McClelland, 2023; Liu et al., 2023; Ramesh et al., 2023; Wang et al., 2024; Khona et al., 2024; Kobayashi et al., 2024). This literature has probed [e.g., via self-attention head visualization (Kovaleva et al., 2019)], the representations that are learned by Transformers when trained on synthetic data sets. These experiments offer a controlled setting where the underlying structure of the task is exactly known and can be manipulated, enabling the identification of local and global attention patterns (Zhang et al., 2022), as well as the ability of shallow Transformers to learn shortcuts to problems that seemingly require recurrence (Liu et al., 2023). Additionally, this work has identified cases where Transformers have the expressiveness to perform compositional reasoning, but fail to do so (Kobayashi et al., 2024). Recent work has performed precise characterization of when shortcut solutions arise in a simplified task and Transformer model Kawata et al. (2025). Our work is motivated by these identified successes and failures in compositional reasoning, as we aim to understand how they impact the ability of Transformers in CL.

## 3 Task

Studying if and how Transformers are able to leverage repeated structure to perform continual compositional reasoning requires a task that can be decomposed into several subcomponents with similar features. To do this in a controlled setting, we turn to a synthetic task, Learning Equality and Group Operations (LEGO) (Zhang et al., 2022). Synthetic tasks have become increasingly recognized as an essential platform with which to study the emergent properties of Transformers that may underlie their success and failure on massive and complex data sets (Geiger et al., 2021; Zhang et al., 2022; Allen-Zhu & Li, 2023; Li & McClelland, 2023; Liu et al., 2023; Ramesh et al., 2023; Wang et al., 2024; Khona et al., 2024). The LEGO task, being group theoretic by nature, has connections to a rich theoretical framework that underlies many of the applied problems Transformers are used on. Recent work has used group theory to achieve an understanding of Transformer performance on a wide range of tasks (Liu et al., 2023). Additionally, LEGO can be viewed as directed graph traversal, connecting LEGO to a large literature on knowledge graph reasoning tasks (Bordes et al., 2013; Ji et al., 2022). LEGO's simplified nature also enabled the identification of local and global attention patterns, that proved to be powerful for solving the task (Zhang et al., 2022).

## 3.1 Learning Equality and Group Operations (LEGO)

Let $G$ be a group, defined by a set of elements $X = \{x_1, ..., x_N\}$ and an operation $*$. A standard example of a group that plays a major role in many domains is the symmetry group of order $k$, $D_k$. One such a symmetry group is $D_3$, which represents the symmetries of a triangle. This is schematically illustrated in Fig. 1A. In this case, there are 6 group elements, each of which correspond to one configuration of a triangle with labeled angles (or equivalently, labeled edges). Each configuration can also be thought of as an action (e.g., rotation by 120°) from a "reference" triangle.

The LEGO task considers a sequence of $T$ symbols, $\{a_t\}_{t=1}^T$, sampled from a library of symbols, $a_t \in A$, where $|A| = M > T$. A sequence of $T$ group elements, $\{x_t\}_{t=1}^T$, $x_t \in X$, is also sampled. From the $a_t$ and $x_t$, a target sequence is generated via the recurrence relation, $a_1 = x_1$ and $a_t = a_{t-1} * x_t$, for $t = 2, ..., T$. We refer to each these $a_t = a_{t-1} * x_t$ units as a "clause". To solve the LEGO task, the correct group element must be assigned to the symbol of each clause. That is, the Transformer must properly identify $x_i \in X$ such that $a_t = x_i$. An example of this, in the context of $D_3$, is presented in Fig. 1B. Note that this iterative

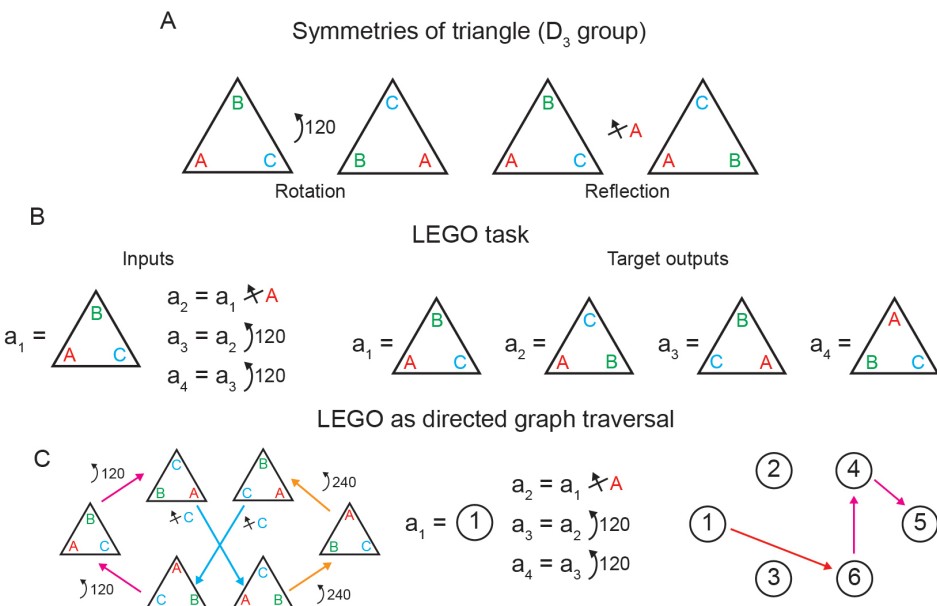

Figure 1: **Schematic illustration of LEGO task.** (A) Illustration of some of the symmetries of the triangle, such as rotation by $120°$ degrees and reflection over one of the three axes. Each of these symmetries is an element of the $D_3$ symmetry group. (B) Illustration of the LEGO task, when applied to group elements of $D_3$. An example input sequence is shown on the left and the target outputs are shown on the right. (C) An illustration of how $D_3$ can be viewed as a direct graph is shown on the left and an example of the LEGO task, when viewed as a directed graph traversal problem, is shown on the right.

composition of $a_{t-1}$ with $x_t$, which is necessary to solve all of the clauses, is what enables the LEGO task to probe a given Transformer model's ability to perform compositional reasoning.

To increase the complexity of the task, and to encourage the convergence to non-trivial solutions, the sequence is presented all at once to the Transformer in a shuffled order. Increasing the length of the target sequence, $T$, can additionally add complexity to the task. For more details on the LEGO task, see Zhang et al. (2022).

Another way to view the LEGO task is as a traversal through a directed graph (Fig. 1C, left). In particular, each group element, $x_i$, can be viewed as nodes in a directed graph, with edges, $e_{i,j} = x_k \in X$ such that $x_i * x_k = x_j$. Solving the LEGO task then amounts to traversing through the graph (Fig. 1C, right), when given a starting location ($a_1 = x_1$) and sequence of edges ($x_t$). This perspective aligns the LEGO task with other recent work on studying the ability of Transformers to navigate through knowledge (and other abstract) graphs (Khona et al., 2024).

In the original LEGO work, Zhang et al. (2022) focused on the $D_2$ group (i.e., the binary flip-flop task), although they also provided some experiments on $D_3$. Sequences of length $T = 6$ were trained on and sequences of length $T = 12$ were tested on. The increased length of the test sequences enabled an understanding of how well the Transformer models were able to generalize to longer sequences. Such long sequences were found to be challenging for BERT and ALBERT models, especially in the case of $D_3$. We therefore reduced the number of elements in the sequence for both training, $T = 4$, and testing, $T = 6$.

### 3.2 Continual LEGO

The $D_3$ group can be decomposed into subcomponents (i.e., subgroups) that have the same structure. Let $x_1$ denote the identity group element. That is, $x_1 * x_i = x_i$, for all $x_i \in X$. Then, there are two group elements $x_2, x_3 \in X$, such that $x_2 * x_2 * x_2 = x_1$ and $x_3 * x_3 * x_3 = x_1$. Namely, if you rotate a triangle $120°$ or $240°$ three times, you get back to the same configuration of the triangle you started with. Similarly, there are three group elements $x_4, x_5, x_6 \in X$, such that $x_4 * x_4 = x_1$, $x_5 * x_5 = x_1$, and $x_6 * x_6 = x_1$. In

particular, if you reflect a triangle twice along the same axis, you get the same configuration of the triangle you started with.

Given the similarity between the group elements $x_4$, $x_5$, and $x_6$, which all have the binary flip-flop like structure, we might expect that a Transformer trained to perform the LEGO task, when restricted to just $x_1$ and $x_4$, may be able to perform well when then subsequently applied to the LEGO task, restricted to just $x_1$ and $x_5$ (or $x_1$ and $x_6$). For this reason, we create three "flip-flop experiences" (Fig. 2A), defined by $x_4$, $x_5$, and $x_6$ (for example sequences from each experience, see Appendix A). We sequentially train BERT and ALBERT models on each of these experiences, probing how well the models are able to generalize to new experiences and how well the models are able to retain their performance on past experiences. For the remainder of the paper, this is what is meant by "continual LEGO".

We note that, while we focus in this work primarily on these three flip-flop experiences, the continual LEGO framework – namely, defining experiences based on subgroups of the larger group – allows for generalization to other experimental set-ups. By considering larger underlying groups (e.g., $D_4$), there exist more subgroups with similar structure that could be turned into different experiences. Thus, we view continual LEGO as an ideal playground for studying continual compositional reasoning.

## 4 Results

### 4.1 BERT and ALBERT performance on continual LEGO

We sequentially train vanilla versions of BERT and ALBERT models, from random initialization, for 100 epochs on flip-flop experiences 1–3 (300 epochs total; see Appendix B, for more training details). These experiences have the same subgroup properties, namely they consist of a single cycle (Fig. 2A). To assess the ability of Transformer models to perform CL in a strict setting, we do not provide any inputs from previous experiences. That is, the Transformer models only have a 100 epochs of exposure to a given experience and then do not see that examples from that experience again.

We find that both BERT and ALBERT models are able to perform well on experience 1, with accuracy on $a_4$ (the length of the sequence used for training) reaching 100% accuracy after 100 epochs (Fig. 2B, C blue solid line, left). However, only ALBERT is able to generalize to longer sequences, as BERT achieves accuracy at around chance level (50%) for $a_5$ and $a_6$ (Fig. 2B, C orange and green dashed lines, left). Similar differences between BERT and ALBERT models were found by Zhang et al. (2022). Upon exposure to the next two flip-flop experience, ALBERT is able to achieve increasingly high accuracy for $a_4$ to $a_6$ (Fig. 2C). This is evidence of forward transfer (New et al., 2022; Baker et al., 2023), suggesting that ALBERT is able to leverage its learned representations to more efficiently achieve high performance on new experiences. Intriguingly, this forward transfer was found to be weaker in BERT, which required more epochs to achieve its asymptotic accuracy of $a_4$ on later experiences (Fig. 2B, middle and right). In addition, one of the four seeds trained was not able to learn $a_4$ for the new experiences. This may be due to the previous discovery that BERT models learn shortcut solutions when trained on the original LEGO task (Zhang et al., 2022). For both ALBERT and BERT, we find that, upon exposure to a new flip-flop experience, the performance on previous flip-flop experiences goes to nearly 0%, demonstrating complete catastrophic forgetting (Fig. 2B, C).

### 4.2 Transformer architecture differentially affects BERT and ALBERT performance on continual LEGO

The BERT and ALBERT models used in the original LEGO work and our analysis above (Fig. 2) are large, with 12 layers and 12 attention heads per layer. This makes it challenging to systematically understand the representations that emerge. Additionally, given the relative simplicity of the tasks involved, such large architectures may be "overkill" and could negatively impact performance and generalization.

To understand how architectural choices shape the ability of BERT and ALBERT models to learn the continual LEGO task, as well as to identify a minimal model that is able to solve the task that we can analyze in more detail, we train BERT and ALBERT models with varying number of hidden layers and

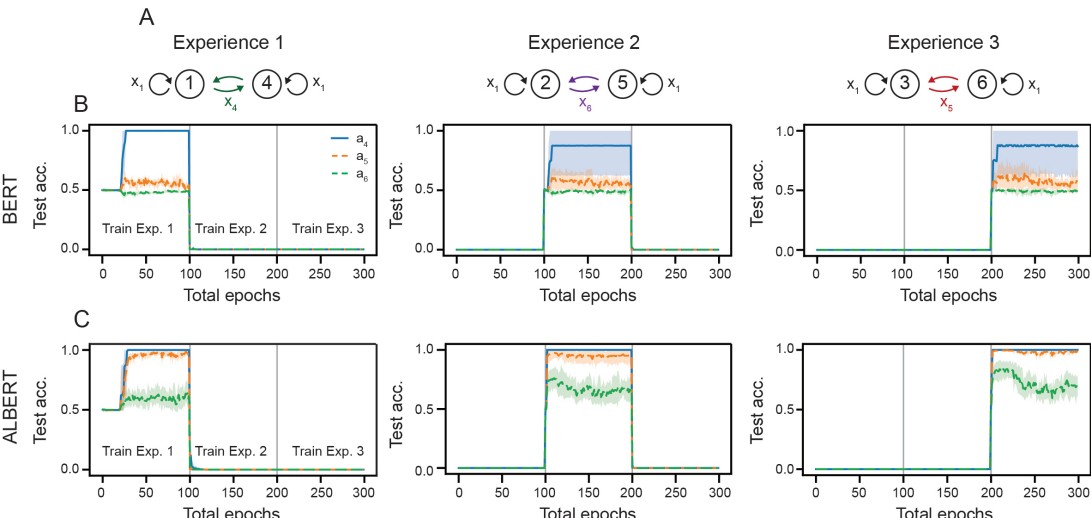

Figure 2: **BERT and ALBERT performance on continual LEGO.** (A) Schematic illustration of the three flip-flop experiences used in the continual LEGO experiments. (B)–(C) The performance of full BERT and ALBERT models, trained from scratch, on the three flip-flop experiences. Left column, accuracy on test sequences of experience 1. Middle column, accuracy on test sequences from experience 2. Right column, accuracy on experience 3. Shaded area denotes 95% confidence interval.

attention heads per layer. For each model, we evaluate the CL performance by computing its task accuracy, generalization accuracy, forward transfer, and performance maintenance (see Appendix C, for details on metrics).

For BERT models, we find that a network with as few as two-hidden layers can learn to perform the current flip-flop experience quite well, if a sufficient number of attention heads are present (Fig. 3A, lower left). This is in-line with previous work showing that Transformers can learn shortcuts to recursive problems Liu et al. (2023). However, for especially large BERT models (12 hidden layers), performance on the current flip-flop experience can degrade (Fig. 3A, lower right), as was shown in Fig. 2B), suggesting an instability in training dynamics. Despite the high test accuracy on variables up to the length of the training sequence ($a_4$), BERT models are generally unable to generalize to perform well on variables beyond that length (e.g., $a_5$) (Fig. 3B). Similarly, while a number of BERT architectures demonstrate forward transfer (Fig. 3C), with the performance of $a_4$ more quickly reaching a high accuracy on flip-flop experience 2 than flip-flop experience 1, there is not a clear relationship with number of hidden layers and attention heads per layer. Across all architectures, there is a complete lack of performance maintenance (Fig. 3D), demonstrating catastrophic forgetting.

For ALBERT models, we find that a network needs at least 4 layers in order to perform well on $a_4$ (Fig. 3E). If at least 4 layers are present, then all architectures (even the largest ones) achieve perfect performance, suggesting stabler training dynamics, relative to BERT. We also find that ALBERT models can achieve good generalization, having nearly perfect performance on $a_5$ as long as they have at least 6 layers (Fig. 3F). In-line with both of these results, we find that forward transfer is largely present as long as there are at least 4 layers (Fig. 3G). However, as with BERT models, we find that nearly all architectural choices for ALBERT lead to a complete inability to maintain performance (Fig. 3H).

Collectively, these results demonstrate that architecture differentially affects the performance capabilities of BERT and ALBERT models on continual LEGO. In particular, while there is a clear relationship between ALBERT's task accuracy, generalization accuracy, and forward transfer with the number of hidden layers (namely, that increasing model capacity leads to great performance), there is less of a clear relationship for BERT models. This suggests that BERT and ALBERT learn and leverage fundamentally different computations.

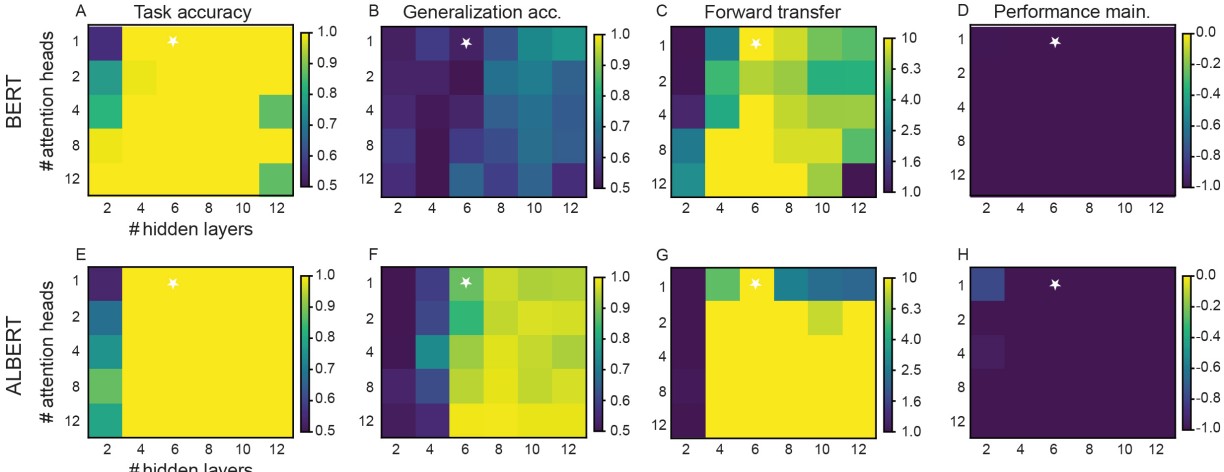

Figure 3: **Transformer architecture differentially affects BERT and ALBERT performance on continual LEGO.** (A) Task accuracy, (B) generalization accuracy, (C) forward transfer, and (D) performance maintenance, as a function of number of hidden layers and attention heads in BERT models. (E)–(H) same as (A)–(D), but for ALBERT models. White star denotes minimal model investigated in more detail. (A)–(H) Metrics are computed over 4 independent seeds. For (C) and (G), the color scale is $\log_{10}$. See Appendix C for details on the CL metrics.

### 4.3 Minimal ALBERT model learns general solution to continual LEGO, while minimal BERT model learns shortcut solution

To better understand these different computations that emerge, we consider a minimal architectural choice for BERT and ALBERT models that achieves high task accuracy, while additionally reflecting the model family's generalization, forward transfer, and performance maintenance. In particular, we chose to investigate minimal BERT and ALBERT models that have 6 hidden layers and 1 attention head, per layer (Fig. 3, white stars). Because these minimal models contain only a single attention head per layer, we can more easily dissect the computations that are learned by analyzing the attention patterns.

We find that, for the minimal ALBERT model, the attention patterns learned after training on flip-flop experience 1 (Fig. 4A) have mechanistic signatures of performing an algorithmic `For` loop to perform compositional reasoning. This can be seen in Fig. 4A, where the calculated self-attention for the tokens in a given clause predominantly attend to tokens in the preceding clause. This dominant preceding clause attention would be expected in an algorithmic `For` loop-esque solution which systematically updates the element estimate for a variable based on the previous clause. This same general trend can be observed after training on flip-flop experience 2 (Fig. 4B), suggesting that this learned computational strategy is being leveraged across experiences. This could be the driver of the strong forward transfer we observe for ALBERT models (Fig. 3G). When summarized between model layers, we find that this signature is pronounced in the later ALBERT layers and increases on exposure to a new flip-flop experience (Fig. 4C, compare blue solid and dashed lines). In contrast, we find that evidence for this kind of computation is weaker in BERT models, where it decreases in later layers (Fig. 4C, compare blue and green lines).

Probing the attention patterns learned by BERT, we find mechanistic signatures of a shortcut solution. This can be seen in Fig. 4D, where instead of the calculated attention being distributed to tokens in the previous clause, the self-attention is isolated to a single shared input token in the first clause across all output tokens. This mechanistic signature of shortcut learning is predominantly seen in later BERT layers (Fig. 4E, green lines) and is not seen in ALBERT models. This shortcut signature persists in BERT, albeit at a slightly reduced level, after exposure to a new flip-flop experience (Fig. 4E, green lines). This could be the driver of the reduced forward transfer performance we observe in BERT models (Fig. 3C), as well as the poor generalization accuracy (Fig. 3B).

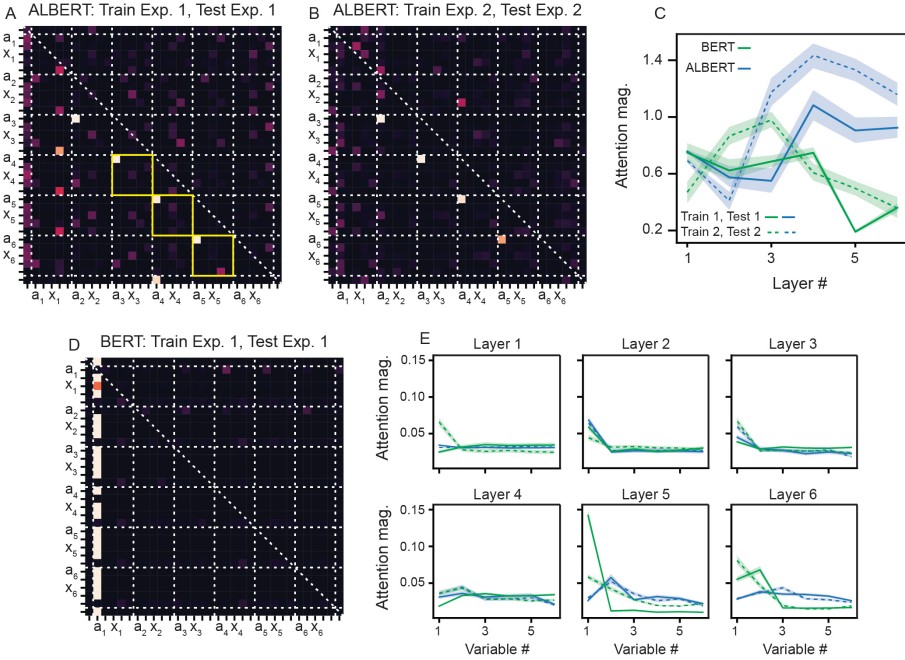

Figure 4: **Attention patterns suggest minimal ALBERT model learns general solution to continual LEGO, while minimal BERT model learns shortcut solution.** (A) Attention pattern for ALBERT minimal model in layer 4, trained on flip-flop experience 1, for an example sequence from flip-flop experience 1. (B) Attention pattern for minimal ALBERT model in layer 4, trained on flip-flop experiences 1 and 2, for example sequence from flip-flop experience 2. (C) Attention patterns averaged over multiple sequences in off-diagonal clauses [yellow squares in (A)], across each layer, for minimal ALBERT and BERT models. Solid blue and green lines are attention computed on models trained on flip-flop experience 1, evaluated on sequences from flip-flop experience 1, and dashed blue and green lines are attention computed on models trained on flip-flop experience 1 and 2, evaluated on sequences from flip-flop experience 2. (D) Same as (A), but for minimal BERT model and layer 5. (E) Attention patterns averaged over multiple sequences for minimal ALBERT and BERT models grouped by input clause [dashed-separated columns in (A), (B), and (D)]. Solid blue and green lines are attention computed on models trained on flip-flop experience 1, with inputs from flip-flop experience 1, and dashed blue and green lines are attention computed on models trained on flip-flop experience 1 and 2, with inputs from flip-flop experience 2. (C) and (E) curves are computed over 4 independently trained seeds and 50 inputs. Shaded area is 95% confidence interval.

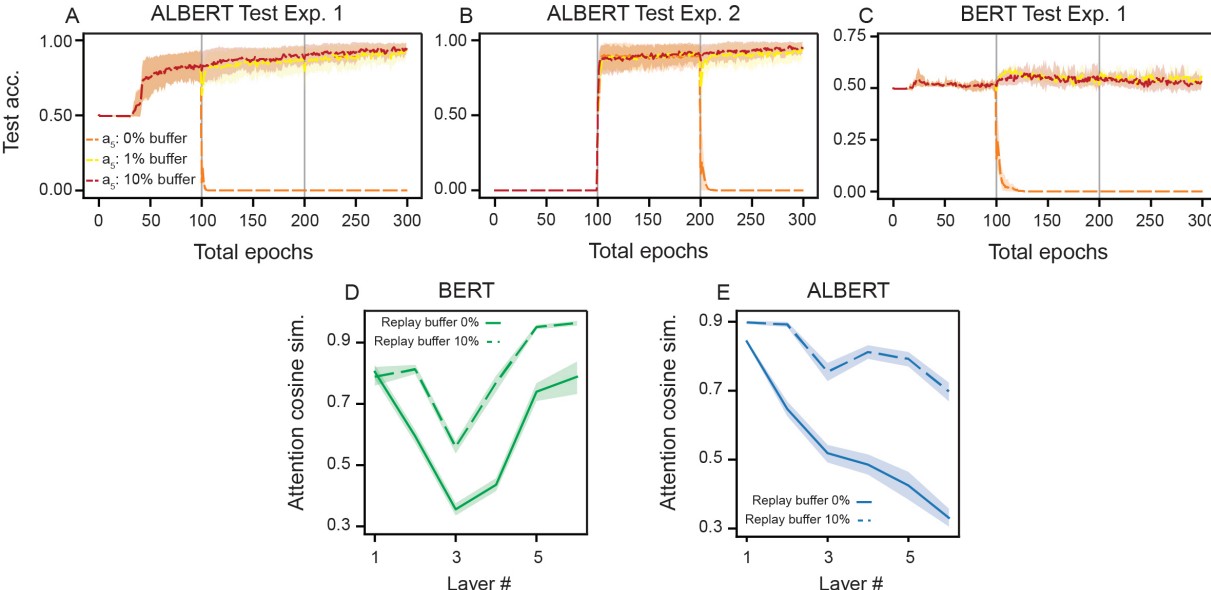

Figure 5: **Replay buffer mitigates catastrophic forgetting, while maintaining forward transfer in minimal BERT and ALBERT models.** (A) Test accuracy on $a_5$ of flip-flop experience 1 for a minimal ALBERT model, as a function of training, for different amounts of replay. (B) Same as (A), but for performance computed on flip-flop experience 2. (C) Same as (A), but for performance of minimal BERT model. (D) Cosine similarity of minimal BERT model flattened attention patterns, computed on sequences from flip-flop experience 1, after training on just flip-flop experience 1 or training on both flip-flop experiences 1 and 2. Solid line denotes minimal BERT models trained with no replay buffer, and dashed line denotes minimal BERT models trained with a 10% replay buffer. (E) Same as (D), but for minimal ALBERT model. Shaded area in all plots is 95% confidence interval.

## 4.4 Incorporation of a replay buffer mitigates catastrophic forgetting in ALBERT and BERT models

Given the significant catastrophic forgetting exhibited by both BERT and ALBERT models (Fig. 3D, H), we sought to identify a method for mitigating this deterioration. As use of replays buffers have exhibited broad success in the CL literature, while being simple to implement, we hypothesize that they may be a useful approach for reducing catastrophic forgetting in continual LEGO. However, it is not clear *a priori*, whether a replay buffer would negatively impact forward transfer.

We find that keeping as little as 1% of training data from past experiences in a replay buffer and randomly including these past examples with examples from the current experience leads to a near complete mitigation of catastrophic forgetting, while leaving forward transfer and generalization intact for minimal BERT and ALBERT models (Fig. 5A–C, yellow lines). Looking at the cosine similarity of the attention patterns, computed on sequences from flip-flop experience 1, before and after training on flip-flop experience 2, we see that the introduction of a replay buffer improves the maintenance of attention patterns when new experiences are introduced (Fig. 5D, E). This suggests that, in addition to improving the strength with which past experiences are encoded in the Transformer models, replay buffers may help solidify the self-attention weights.

## 4.5 Training on incrementally combined experiences can enable greater continual compositional reasoning in minimal ALBERT, but not BERT, models

A major goal of continual compositional reasoning is being able to apply reasoning that integrates between experiences. To characterize the extent to which ALBERT and BERT models are able to perform such reasoning, we develop a new set of experiences that we refer to as "compositional flip-flop" experiences (Fig. 6A, top row). In particular, each experience has the same group structure as before (i.e., contains a cycle with two elements), but each experience additionally shares one of the same elements. This enables us to

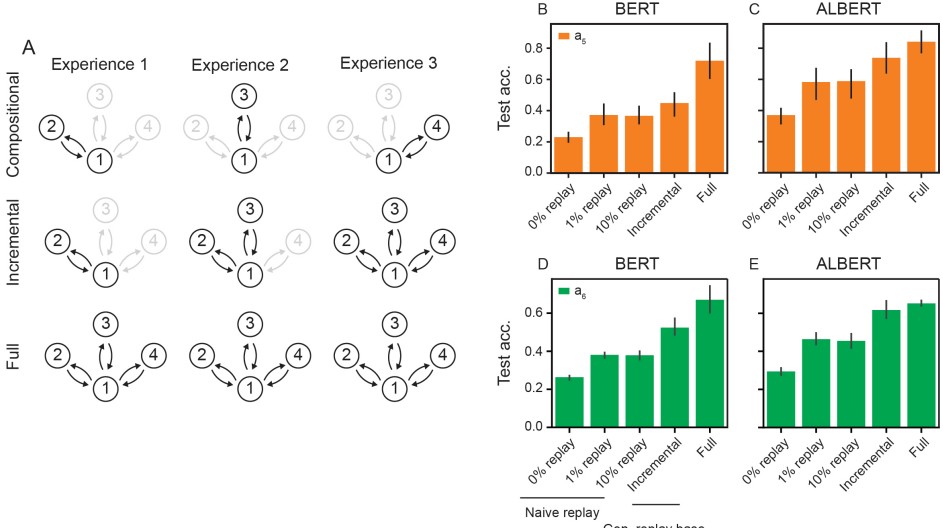

Figure 6: **Naive replay does not enable compositional performance, but training on incrementally combined experiences rescues performance drop.** (A) Schematic illustration of compositional, incremental, and full flip-flop experiences. (B)–(C) Performance on $a_5$ of sequences from the full task for the minimal BERT and ALBERT models trained with different replay buffers and on different tasks. (D)–(E) Performance on $a_6$ of sequences from the full task for the minimal BERT and ALBERT models trained with different replay buffers and on different tasks. Error bars are standard deviation.

identify how well the trained minimal models do on input sequences that come from the "full" task (Fig. 6A, bottom row), which includes sequences that interpolate across different experiences. Good performance on these compositional flip-flop experiences would provide strong evidence that the minimal models learn to compose across experiences.

We find that, with no replay, the minimal ALBERT model has an accuracy of a little better than chance level ($\approx 35\%$) for $a_5$ (Fig. 6C). Incorporation of a replay buffer increases the test accuracy to nearly 60%, a significant increase, although one that is nonetheless around 25% less than what is achieved when the minimal ALBERT model is trained on the full task (85% accuracy). This demonstrates that the naïve replay strategy, which saves examples from each experience separately, is not sufficient to enable this more complete form of compositional which composes across experiences. Given the presence of shortcut solutions Liu et al. (2023), we imagine that replay in this way is not sufficient to link components of tasks which were experienced separately. Similar results are seen for the BERT minimal model (Fig. 6B) and for $a_6$ (Fig. 6D–E).

Recent work in continual learning has led to the development of generative replay strategies (Robins, 1995; Shin et al., 2017; Van de Ven et al., 2020), where – instead of individual examples being saved – a generative model of the underlying input distribution is learned. We hypothesize that such a generative replay mechanism could enable enhanced compositional reasoning. To demonstrate the potential of generative replay, we consider a training strategy where past experiences are stitched together with new experiences ("incremental flip-flop – Fig. 6A, middle). This serves as an upper-bound to the potential success a generative replay strategy might have, given that it trains using sequences that are combined across different experiences. We find that training on the incremental flip-flop task leads to an ALBERT minimal model whose performance is similar to what is achieved when training on the full task (Fig. 6C, E). While the last experience is the full task, we note that training on sub-components of the full task first (i.e., flip-flop experience 1) could lead to representations being learned that prevent good learning on the full task. Indeed, we find that use of this incremental training strategy does not lead to the minimal BERT model being able to perform as well as the full model (Fig. 6B, D). Thus, we believe this result is non-trivial and demonstrates that generative replay may be a particularly fruitful direction for improving the ability of continual compositional reasoning in ALBERT.

## 5    Discussion

Continual learning remains a fundamentally challenging and open area in machine learning. While Transformer models have found remarkable success in compositional reasoning (Geiger et al., 2021; Allen-Zhu & Li, 2023; Li & McClelland, 2023; Liu et al., 2023; Ramesh et al., 2023; Wang et al., 2024; Khona et al., 2024; Kobayashi et al., 2024), and have exhibited the ability to learn local and global attention patterns (Zhang et al., 2022), little is known about their ability to compose across multiple experiences. To begin to address this gap, we investigated the capabilities of feedforward and recurrent Transformer models in simplified and controlled synthetic continual compositional reasoning task. Expanding the LEGO framework to include CL tasks enabled us to explore this question in a controlled setting.

The original LEGO work by Zhang et al. (2022) found that recurrent ALBERT models were better able to generalize to sequences of test length longer than the training length, as compared to feedforward BERT models. This was attributed to the putative `For` loop-esque like structure the recurrent architecture of ALBERT enables. Performing detailed analysis of the attention patterns that emerge in trained ALBERT models, we find evidence of such `For` loop-esque computations (Fig. 4), providing greater support to the hypothesized computational properties of recurrent Transformers. However, further probing is needed to causually show that the attention patterns are truly performing a `For` loop.

When applying BERT and ALBERT models to our continual LEGO task, we find this difference in recurrent versus feedforward architecture becomes even more important in shaping the ability of the models. In particular, while the full ALBERT model is able to increase its speed with which it learns each new related experience, demonstrating forward transfer, the full BERT model can fail to perform beyond chance level performance on all experiences after the first (Fig. 2). Closer examination of the relationship between model size and model performance on continual LEGO demonstrates that ALBERT, but not BERT, models see an increase in performance as model capacity increases. Despite this beneficial property, both the standard ALBERT and BERT models exhibit complete catastrophic forgetting (Fig. 2). This is in-line with work on continual learning with ViTs, where there was found to be a bias towards the newest experience (Yu et al., 2021).

One possible explanation for these differences is that ALBERT, by sharing weights across layers, has fewer parameters to optimize, and therefore may bias the network to achieve a more parsimonious and generalizable solution. In the original LEGO work, Zhang et al. (2022) considered this hypothesis and trained a "thin" BERT, which had fewer hidden units so that the total number of parameters approximately matched the ALBERT models. This reduction in model size led to collapse in performance, where above chance performance was not achieved even for the sequence length of the training distribution. Given that the thin BERT failed on the simpler LEGO task considered by Zhang et al. (2022), we therefore conclude that the difference in trainable parameters is unlikely to underlie the differences between the two models, and that a fundamental difference in architecture is more likely to be responsible. In addition, we find that, when performing the sweep over architectural hyper-parameters, none of the smaller BERT models (either with fewer hidden layers or fewer attention heads per hidden layer) showed as high an ability to generalize as the full ALBERT model (Fig. 3B, F).

Analysis of self-attention patterns in a minimal ALBERT model reveals that the structure learned from one experience was maintained when the network was applied to a similar experience (Fig. 4A, B). In particular, we find evidence that the minimal ALBERT model has learned computations that are `For` loop-esque. This suggests a possible mechanism by which forward transfer can occur. Namely, the computation strategy learned on flip-flop experience 1 is general enough to be directly useful to flip-flop experience 2. This is in contrast with ViTs applied to vision based CL tasks, where the self-attention patterns have been found to become less localized with exposure to new classes (Zheng et al., 2023). Instead, in our continual compositional reasoning task, we find evidence of BERT learning computations that are shortcut solution-like (Fig. 4D, E). This provides an explanation for the poor generalization capabilities of BERT to longer sequences and suggests a fundamental challenge to using feedforward architectures on compositional reasoning tasks.

To mitigate the severe catastrophic forgetting we observe in both models, we tried using a replay buffer. We found that saving as little as 1% of the training examples from past experiences was enough to mitigate catastrophic forgetting, while maintaining forward transfer (Fig. 5). This suggests that improving aspects of Transformer continual compositional learning may be simpler than architectural modifications that have been proposed for ViTs Yu et al. (2021); Wang et al. (2022); Zheng et al. (2023). However, we find that our minimal BERT and ALBERT models, using replay, are not able to sufficiently compose data across experiences, a major limitation in the context of CL (Fig. 6). This suggests that, even in the case of ALBERT, which appears to learn a general solution, a non-optimal solution has been converged to. This can mitigated by using a training strategy where data from past and current experiences are stitched together, but only for ALBERT models. This training strategy acts as an upper-bound for benefits we might expect to see from generative replay methods. However, it is not clear how close to this upper-bound generative replay methods can get, suggesting that future work should further explore the potential of generative replay in continual compositional reasoning.

## 5.1 Limitations

The LEGO framework is a small scale, synthetic data set, which exhibits many significant differences from the real-world data to which Transformer models are typically applied. In addition, we restrict ourselves to studying only BERT and ALBERT models to enable a more direct comparison to the previous LEGO work and to maintain the same task and training structure, which would have to be modified to enable the use of autoregressive models. However, the group theoretic nature of the LEGO task is directly connected to underlying symmetries that are present in complex data. Further, BERT and ALBERT models form a canonical baseline that enables our results to be more clearly interpreted.

This work does not systematically investigate CL approaches for continual reasoning, and our results are limited to the application of replay buffers. While replay buffers have been widely used in practice to mitigate catastrophic forgetting and are a broadly applicable CL technique, many other strategies such as progressive networks Rusu et al. (2016), knowledge distillation Gou et al. (2021), meta-plasticity and contextual modulation Kudithipudi et al. (2022). Novel approaches for CL for Transformer architectures are being developed for multimodal tasks Cai & Rostami (2024) and a variety of CL approaches investigated in the context of large language models using transformer architectures Wu et al. (2024); Shi et al. (2024). While this work has established the potential for forward transfer in Transformer architectures for compositional reasoning tasks, future work will be required to investigate the interaction between CL approaches and Transformers architectures to fully characterize this phenomena. Given the unique structure of the proposed compositional reasoning tasks, we believe this will be a promising approach for the community to characterize reasoning tasks in the context of continual learning with transformer models.

## 5.2 Future directions

Recent work has introduced novel architectural elements to enhance and separate the relational reasoning of Transformers (Altabaa & Lafferty, 2024; Altabaa et al., 2024). Future work can explore the potential of these architectures for CL, particularly in relation to compositional reasoning tasks. In particular, strategies which can effectively exploit the compositional aspects of reasoning tasks are of high importance. Furthermore, future work can adapt our framework to enable modern autoregressive Transformer models to be investigated, an important direction to expand the characterization of continual compositional reasoning in transformer architectures.

In addition, it will be critical to expand the notion of compositional reasoning to additional tasks beyond LEGO. While ideal for detailed assessment of transformer performance in continual compositional reasoning tasks, the limited scope of the LEGO tasks limit applicability to large-scale transformer model training. Expanding test cases to conditions such as large knowledge graphs or compositional reasoning grounded in real-world data is a critical step, but will require careful development to maintain the rigor of the group operations formulation of LEGO.

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

## A  Example continual LEGO inputs

As noted in Sec. 3.2, our continual LEGO task was built upon the $D_3$ group. This was represented to the Transformer models as a set of group elements – $X = $ val, rotate, spin, flip, reflect, mirror – and a Cayley table, which defines how different group elements acted on each other. For instance, val was designated as the identity element, such that val $\circ$ $x = x$ for all $x \in X$. rotate and spin are inverses of each other (i.e., rotate $\circ$ spin $=$ spin $\circ$ rotate $=$ val). These correspond to the action of rotating a triangle by 120° and 240°, respectively (Fig. 1). In contrast, flip, reflect, and mirror are their own inverses (i.e., flip $\circ$ flip $=$ reflect $\circ$ reflect $=$ mirror $\circ$ mirror $=$ val). These correspond to the action of reflecting a triangle along each of its three axes (Fig. 1).

Below, we give example input sequences from each of flip-flop experiences 1, 2, and 3. Each experience is comprised of two group elements, $x_i$ and $x_j$, that can be mapped to each other through the action of a third group element, $x_k$. That is $x_i \circ x_k = x_j$ and $x_j \circ x_k = x_i$. We refer to $x_i$ and $x_j$ as the "elements" of the experience and $x_k$ as the "relation" of the experience. We also include the identity group element, $x_1$, as a relation for each experience. In this way, each experience is analogous to the original binary flip-flop task Zhang et al. (2022).

For sake of clarity, we will present them as ordered, although – as noted in Sec. 3.1 – the actual inputs to the Transformer models were shuffled, to make the task more complex (Zhang et al., 2022). Similarly, we also present the symbols that are used in an ordered way, but the actual task randomly selects symbols.

**Experience 1**

Elements: {spin, mirror}

Relations: {val, reflect}

Example input sequence: a $=$ spin, b $=$ a $\circ$ val, c $=$ b $\circ$ reflect, d $=$ c $\circ$ reflect.

Example target output: a $=$ spin, b $=$ spin, c $=$ mirror, d $=$ spin.

**Experience 2**

Elements: {rotate, reflect}

Relations: {val, mirror}

Example input sequence: a $=$ rotate, b $=$ a $\circ$ val, c $=$ b $\circ$ mirror, d $=$ c $\circ$ val.

Example target output: a $=$ rotate, b $=$ rotate, c $=$ reflect, d $=$ reflect.

**Experience 3**

Elements: {val, flip}

Relations: $\{\mathtt{val}, \mathtt{flip}\}$

Example input sequence: $\mathtt{a} = \mathtt{val}, \mathtt{b} = \mathtt{a} \circ \mathtt{flip}, \mathtt{c} = \mathtt{b} \circ \mathtt{flip}, \mathtt{d} = \mathtt{c} \circ \mathtt{flip}$.

Example target output: $\mathtt{a} = \mathtt{val}, \mathtt{b} = \mathtt{flip}, \mathtt{c} = \mathtt{val}, \mathtt{d} = \mathtt{flip}$.

## B  Training details

Code for the experiments was generated by modifying the official public LEGO repository, https://github.com/yizhangzzz/transformers-lego. BERT and ALBERT models were trained from random initialization using the HuggingFace `Transformers` package. Training was performed using Adam optimization, with a learning rate of $5 \cdot 10^{-5}$, and a cross-entropy loss. The learning rate was modified using cosine annealing, with a $T_{\max} = 200$. The training and test sets were constructed of $60,000$ and $6,000$ examples, per flip-flop experience. The batch size was set to $500$.

## C  Continual learning metrics

To quantify the performance of BERT and ALBERT models on the continual LEGO task, we make use of four metrics: task accuracy, generalization accuracy, forward transfer, and performance maintenance (New et al., 2022; Baker et al., 2023). We provide more details on these below.

**Notation:** Let $C_j^i(k)$ be the test accuracy of an ALBERT or BERT model on predicting the value of $a_j$, assessed with inputs from flip-flop experience $i$, after training for $k$ epochs. Note that each model is trained for 100 epochs on each flip-flop experience (sequentially). Thus, for the first 100 epochs, the model has only been trained on flip-flop experience 1, and for the first 200 epochs, the model has only been trained on flip-flop experience 1 and 2. Let $\tau_j^i(\alpha) = \min_k \{C_j^i(k) > \alpha | k = 100 \cdot i + 1, ..., 100 \cdot (i+1)\}$. That is, $\tau_j^i(\alpha)$ is the first epoch, when training on experience $i$ that the accuracy gets to be above $\alpha$, where $\alpha \in [0, 1]$.

**Task accuracy:** We define the task accuracy metric as

$$\text{TA} = \frac{1}{10} \sum_{k=291}^{300} C_4^3(k). \tag{1}$$

That is, the task accuracy metric is the accuracy in predicting $a_4$, assessed on flip-flop experience 3, averaged over the last 10 epochs. TA $= 1$ denotes perfect accuracy on predicting the value of $a_4$ and TA $= 0.5$ denotes accuracy that is equivalent to chance.

**Generalization accuracy:** We define the generalization accuracy metric as

$$\text{GA} = \frac{1}{10} \sum_{k=291}^{300} C_5^3(k). \tag{2}$$

That is, the generalization accuracy metric is the accuracy in predicting $a_5$, assessed on flip-flop experience 3, averaged over the last 10 epochs. GA $= 1$ denotes perfect accuracy on predicting the value of $a_5$ and GA $= 0.5$ denotes accuracy that is equivalent to chance.

**Forward transfer:** We define the forward transfer metric as

$$\text{FT} = \frac{\tau_4^1(0.9)}{\tau_4^2(0.9)}. \tag{3}$$

That is, the forward transfer metric is the ratio of the number of epochs it takes the model to learn $a_4$ (up to 90% accuracy) on experience flip-flop experience 1 and the number of epochs it takes the model to difference in predicting $a_4$ on flip-flop experience 2. A FT $= 1$ denotes no forward transfer (as it takes the same number of epochs to learn $a_4$ on flip-flop experience 2 as it did on flip-flop experience 1) and FT $>> 1$ denotes strong forward transfer (as it takes many fewer epochs to learn $a_4$ on flip-flop experience 2, as compared to flip-flop experience 1).

We note that, because each flip-flop experience is effectively the same, the number of epochs it takes for the performance to reach 90% accuracy on flip-flop experience 1 – from random initialization – is (in expectation) the same as it takes for the performance to reach 90% accuracy on flip-flop experience 2 – from random initialization. Therefore, what is being computed in the equation above is equivalent to the number of epochs needed for the network, pretrained on flip-flop experience 1, to reach 90% accuracy on flip-flop experience 2, vs. the number of epochs needed for the network, from random initialization, to reach 90% accuracy on flip-flop experience 2.

**Performance maintenance:** We define the performance maintenance metric as

$$\text{PM} = \frac{1}{10} \cdot \frac{\sum_{k=101}^{110} C_4^1(k) - \sum_{k=1}^{10} C_4^1(k)}{\sum_{k=101}^{110} C_4^1(k) + \sum_{k=1}^{10} C_4^1(k)}. \tag{4}$$

That is, the performance maintenance metric is the difference in predicting $a_4$ (from sequences sampled from flip-flop experience 1) before and after training on flip-flop experience 2. $\text{FT} = 0$ denotes perfect maintenance and $\text{FT} = -1$ denotes no catastrophic forgetting.

