# OpenReview forum: "Shortcut Solutions Learned by Transformers Impair Continual Compositional Reasoning"
_TMLR — Rejected by TMLR_

### Review · Reviewer_pJLh · 2026-03-14

**Summary Of Contributions:**

The paper extends the LEGO benchmark to a continual-learning setting and uses this controlled setup to study how transformer architectures affect compositional reasoning across related experiences. It compares BERT and ALBERT in forward transfer, length generalization, catastrophic forgetting, architectural scaling, replay-based mitigation, and cross-experience composition. The main conclusion seems to be that ALBERT shows stronger forward transfer and better generalization, while BERT appears more susceptible to shortcut-based solutions; at the same time, both struggle at composing information across experiences. The paper also contributes an analysis of attention patterns, suggesting a more loop-like computation in ALBERT and a more brittle shortcut in BERT, together with replay experiments showing partial mitigation of forgetting in the continual setting. Overall, it is an interesting contribution within a narrow synthetic framework, and the conclusions are likely more convincing at the level of this benchmark than as general claims about continual compositional reasoning using transformers.

**Additional Comments:**

I'd love to see the CL-LEGO code as an open-source contribution linked in the paper directly...

**Audience:**

Yes

**Audience Explanation:**

The paper sits at the intersection of several topics that are clearly relevant to TMLR readers: mechanistic understanding of transformers, compositional reasoning, inductive bias, and continual learning. The fact that is it so narrow does reduce its audience to folks specifically interested in synthetic mechanistic studies however.

**Claims And Evidence:**

No

**Claims Explanation:**

The presented evidence is clear and convincing for the benchmark-level empirical findings, but not for the paper's strongest mechanistic interpretation, i.e., the claim that ALBERT learns a For-loop-like solution. That claim appears to rest on correlational evidence from attention patterns rather than on a causal demonstration. This seems fixable, either by softening the wording of the claim or by adding intervention-based evidence such as head ablations, attention patching, or related causal tests.

**Requested Changes:**

- Strengthen or soften the mechanistic claim around the "for-loop" (critical). The current evidence seems suggestive rather than causal: the paper shows attention patterns consistent with a loop-like computation, but repeatedly frames this as "direct evidence" that ALBERT has learned a For-loop solution. This claim would be much stronger with intervention-based evidence, e.g. head ablations, patching.

- Calibrate the scope of the claims to the actual evidence (critical). The empirical results are convincing within the proposed framework/sandbox, but the paper is built around a narrow synthetic setting and an old BERT/ALBERT comparison. Narrow the framing in the abstract and discussion so that the paper is presented primarily as a controlled study of continual compositional reasoning (instead of a broader claim about recurrent versus feedforward transformers).

- Clarify the "generative replay" section. As written, this part is somewhat confusing because the paper discusses a generative replay baseline, but then explains that the actual experiment is an "approximate upper bound" obtained by stitching past and new experiences together.

- Try to broaden the experimental picture (if possible). It would be great to include at least one additional axis of validation: for example, a larger group/task variant, a second compositional setting, or a more modern architecture baseline to confirm findinds.

---

> ### Author Response · Authors · 2026-03-29
> **Rebuttal 1**
>
> We thank the reviewer for their thoughtful and constructive feedback. Below, we address each of their points and reference places in the revised manuscript where changes have been made, which we believe has made our work stronger.
>
> **"Strengthen or soften the mechanistic claim around the "for-loop" (critical). The current evidence seems suggestive rather than causal: the paper shows attention patterns consistent with a loop-like computation, but repeatedly frames this as "direct evidence" that ALBERT has learned a For-loop solution. This claim would be much stronger with intervention-based evidence, e.g. head ablations, patching."**
>
> We agree with the reviewer that our original description of the For loop was too strong. We have reduced the statements regarding this in the Abstract, Introduction, Sec. 4.3, and Discussion of the revised manuscript.
>
> **"Calibrate the scope of the claims to the actual evidence (critical). The empirical results are convincing within the proposed framework/sandbox, but the paper is built around a narrow synthetic setting and an old BERT/ALBERT comparison. Narrow the framing in the abstract and discussion so that the paper is presented primarily as a controlled study of continual compositional reasoning (instead of a broader claim about recurrent versus feedforward transformers)."**
>
> We appreciate the reviewer pointing out that the claims were too broad given the results. We have narrowed our focus in the revised manuscript, particularly in the Introduction and Discussion.
>
> **"Clarify the "generative replay" section. As written, this part is somewhat confusing because the paper discusses a generative replay baseline, but then explains that the actual experiment is an "approximate upper bound" obtained by stitching past and new experiences together."**
>
> We appreciate the reviewer pointing out this confusing explanation of our results. To improve this, we have generally reduced our framing of these results as a “generative replay strategy” in the Abstract, Introduction, Sec. 4.5, and the Discussion. In addition, we have tried to provide more detail on how this approach can be thought of as an upper-bound to generative replay strategies, and what challenges real generative replay strategies might face in the Discussion.
>
> **"Try to broaden the experimental picture (if possible). It would be great to include at least one additional axis of validation: for example, a larger group/task variant, a second compositional setting, or a more modern architecture baseline to confirm findings."**
>
> We agree that our choice in analyzing BERT and ALBERT models is restrictive, particularly given today’s use of autoregressive models. However, to be able to train such autoregressive models on the LEGO task (and, by extension, the continual LEGO task), it would be necessary to modify the training procedure and the way in which the task was structured. We believe this is an important direction to pursue, but would complicate direct comparison of such models with ALBERT and BERT. Therefore, we have decided to stick with just considering BERT and ALBERT. We have added discussion in our revised manuscript of this important limitation in our Abstract, Introduction, and Discussion sections. Additionally, we emphasize that adapting continual LEGO to modern architectures and tasks is an interesting future direction.

---

### Review · Reviewer_W8Qe · 2026-03-17

**Summary Of Contributions:**

The paper studies continual learning in the synthetic reasoning task, "Learning Equality and Group Operations" (LEGO), introduced by prior work [1].
Each task in LEGO consists in resolving the values of a number of variables, where each variable is defined as a function of another variable, or a concrete value.
For example, an input could be $x = 1, y = g(z), y = f(x)$ and the task is to resolve the values of $x$, $y$ and $z$.
Valid functions and values in LEGO tasks are defined over a specific group, e.g. the Dihedral Group $D_3$ of symmetries of triangles:
The functions correspond to the actions of the group (rotations and reflections) and the values correspond to the elements in the group (the 6 different orientations of a triangle).
The paper adapts this task for a continual learning setting by sequentially considering tasks defined over subgroups of $D_3$.
Like in [1], this paper compares BERT models with ALBERT models, a variant of BERT where the weights for each layer are shared.
Networks are continually trained on 3 tasks ("experiences") consisting of 4 variables to be resolved, and like in [1], length extrapolation is studied during evaluation by considering tasks consisting of 4, 5 and 6 variables.
The paper finds that continually trained networks (1) struggle to generalize from the training to the test set (2) achieve limited length extrapolation and (3) catastrophically forget previous tasks unless a replay buffer is added.
ALBERT models achieve better generalization to the test set which is reflected in their attention patterns that show signs of resolving the chain of variables as required by the task.

[1] Unveiling Transformers with LEGO: a synthetic reasoning task, Zhang et al., ICLR 2023

**Audience:**

No

**Audience Explanation:**

The paper studies a slight modification of an existing task and uses existing methods (replay buffer, BERT/ALBERTA).
For it to be of interest, it would have to present a compelling analysis in the continual learning setting but unfortunately I do not think this is the case.

**Broader Impact Concerns:**

I see no specific concerns for the broader impact of this paper.

**Claims And Evidence:**

No

**Claims Explanation:**

Overall, I find the claims made in the submission confusing at best with a large overlap to prior work. I will go through the main contributions listed in section 1 as well as the title one by one in the following to substantiate this.

> Shortcut Solutions Learned by Transformers Impair Continual Compositional Reasoning

The title of the paper introduces two terms that are used throughout the paper as the framing: "shortcut solutions" and "compositional reasoning".

The term "shortcut solutions" is used to refer to "solutions that are non-generalizable", i.e. solutions that do not generalize from the training to the test set despite achieving a low loss on the training set.
I find this term not really useful as an explanatory device in the context of this paper beyond the notion of overfitting since there isn't any substantial analysis in what way such solutions fail.
The fact that overfitting is observed here might also not be surprising given that models are trained for 100 epochs, i.e. repeating the same specific tasks many times rather than sampling fresh tasks "online" which would be equally possible in this synthetic setting.

Throughout the paper the term "compositional reasoning" is used but it is unclear what distinguishes compositional from non-compositional reasoning?
The synthetic reasoning task that this paper is based on [1] simply calls this task a synthetic reasoning task and the introduction does unfortunately not help to clarify this part either.

> Our work provides the first in-depth analysis of how feedforward and recurrent Transformer models perform on continual compositional reasoning.

If we remove the in my opinion vacuous term "compositional", this statement is simply too broad.
The paper considers a particular synthetic reasoning task that is likely not indicative of general reasoning in Transformers.

> We expand the synthetic compositional reasoning LEGO task (Zhang et al., 2022) to enable a systematic investigation of CL capabilities of Transformer models.

The way in which the previously introduced LEGO task framework is split into subtasks via subgroups for continual learning is neat since the structure of each subtask is similar to a task on the permutation group of two elements while all subtasks together combine to the permutation group of three elements.

> We find that architectural choices (e.g., number of attention heads, number of hidden layers) differentially affect the generalization accuracy and strength of forward transfer for BERT and ALBERT models, with BERT models demonstrating inconsistent performance on CL, as model size is in-
creased.

I am missing a clear hypothesis or meaningful experimental finding with respect to the search over architectural hyperparameters beyond one configuration working better than another.
What I find particularly puzzling is that the configuration chosen after conducting this hyperparameter search is subobtimal in Figure 3 (selecting a single attention head), skewing the remainder of the results.
The paper argues that this is to enable the analysis of the attention patterns but I do not think that is sufficient ground.
It is certainly possible to analyse attention patterns for multiple heads.

> We identify signatures of shortcut and algorithmic For loop solutions in BERT and ALBERT models, respectively, providing a mechanistic explanation of their different performance on the continual LEGO task.

The variable resolution chain is reflected in the attention patterns of ALBERT models which a generalizing solution must in some form accomplish.
A similar finding was presented in [1].
The analysis of the shortcut solution is limited to the observation that its attention pattern is not displaying this pattern, instead overly attending to a single, uninformative token.

> We demonstrate that both families of models fail to perform well on a continual LEGO setting that requires composition across experiences, a failure that we suggest can be rescued with a generative replay strategy in ALBERT models, but not BERT models, where a shortcut solution has become entrenched.

The paper refers to a "generative replay strategy" but there is no generative model being used.
The paper states "To demonstrate the potential of generative replay, we train the minimal models in a setting where past experiences are stitched together with new experiences (“incremental flip-flop – Fig. 6A, middle). This serves as an approximate upper bound to the potential success a generative replay strategy might have."
What is actually being done is the usage of a simple replay buffer that has been extensively used in the continual learning literature.
The results do not go beyond the finding that a replay buffer works in this setting.

**Requested Changes:**

I do not want to give the misleading impression that changing specific parts of the paper would suffice to resolve the substantial concerns I have raised above and I am thus refraining from making such requests.

---

> ### Author Response · Authors · 2026-03-29
> **Rebuttal 1**
>
> We thank the reviewer for their detailed and critical feedback. Below, we address each of their points and reference places in the revised manuscript where changes have been made, which we believe has made our work stronger.
>
> **"The term "shortcut solutions" is used to refer to "solutions that are non-generalizable", i.e. solutions that do not generalize from the training to the test set despite achieving a low loss on the training set. I find this term not really useful as an explanatory device in the context of this paper beyond the notion of overfitting since there isn't any substantial analysis in what way such solutions fail. The fact that overfitting is observed here might also not be surprising given that models are trained for 100 epochs, i.e. repeating the same specific tasks many times rather than sampling fresh tasks "online" which would be equally possible in this synthetic setting."**
>
> We appreciate the reviewer pointing out confusion regarding our choice of terminology. Shortcut solutions have been studied in several contexts regarding Transformer models trained on sequential tasks [Liu et al., 2023 ICLR; Kawata et al., 2025 NeurIPS]. This work has found that such models can learn to perform the task in highly specialized ways. For instance, in the original LEGO paper (Zhang et al., 2022 arXiv) it was found that BERT’s performance on the last element in the chain could be high, even if performance on other elements of the chain was low. This was conjectured to be due to the fact that - by counting the number of times a certain relationship was used, it was possible to immediately figure out what the last element of the chain should be. That this is learned, and not a more general solution, was considered to be evidence of a “short-cut” (in addition to analysis studying the attention patterns in BERT).
>
> We similarly find BERT models learn very specific attention heads and that BERT does not learn solutions that generalize well to longer sequences and new experiences, despite the fact that ALBERT - trained for the same number of epochs on the same task - learns more general attention heads and can generalize to longer sequences and new experiences. As Zhang et al. found that BERT models trained with the same number of parameters as ALBERT still learn a less generalizable solution (their Fig. 17), we believe this is not a trivial problem of over-fitting, but rather a specific property of BERT that leads to short-cut solutions to be learned more readily than generalizable solutions. In recent work by Kawata et al., 2025 NeurIPS (which we now cite in our revised manuscript), a precise characterization of when the “general” or shortcut solution (for a specific simplified task and Transformer model) would be learned was performed.
>
> We note that, because we used 26 different symbols (as was done in the original LEGO paper), there are 26^6 * 2^4 possible training sequences. Thus, even though we train for  many iterations, we are not repeating the exact same sequences, but slightly different ones across training. This prevents direct memorization.
>
> **"Throughout the paper the term "compositional reasoning" is used but it is unclear what distinguishes compositional from non-compositional reasoning? The synthetic reasoning task that this paper is based on [1] simply calls this task a synthetic reasoning task and the introduction does unfortunately not help to clarify this part either."**
>
> We again thank the reviewer for pointing out an area where there is a lack of clarity. Compositional reasoning tasks have recently been widely used to study properties of Transformer models [e.g., Liu et al., 2023 ICLR; Khona et al., 2024 ICML]. In particular, these tasks generate sequences where the model must iteratively compose specific actions with the resulting state of the system. For LEGO, this comes from the recurrence relation a_t = a_t - 1 * x_t, whereby the current value a_t - 1 is composed with the action of the group element x_t. A non-compositional task would be one where, for each a_t, a specific relationship must be learned, independent of any action on the current state. For instance, a_1 = x_1, a_2 = x_3, a_3 = x_5, … To make this point more clear, we have added discussion on this in the Introduction and Sec. 3.1 of the revised manuscript.
>
> **"If we remove the in my opinion vacuous term "compositional", this statement is simply too broad. The paper considers a particular synthetic reasoning task that is likely not indicative of general reasoning in Transformers."**
>
> We agree that our original claims were overly strong. In the revised manuscript, we have reduced them to reflect more accurately the specific task and models we study, particularly in the Abstract, Introduction, and Discussion.

---

> > ### Author Response · Authors · 2026-03-29
> > **Rebuttal 2**
> >
> > **"I am missing a clear hypothesis or meaningful experimental finding with respect to the search over architectural hyperparameters beyond one configuration working better than another. What I find particularly puzzling is that the configuration chosen after conducting this hyperparameter search is subobtimal in Figure 3 (selecting a single attention head), skewing the remainder of the results. The paper argues that this is to enable the analysis of the attention patterns but I do not think that is sufficient ground. It is certainly possible to analyse attention patterns for multiple heads."**
> >
> > From the search over architectural hyperparameters, we concluded two meaningful things. First, in the original LEGO paper, only the full BERT and associated ALBERT models were considered. Given that these are very large architectures, and the task relatively simple, it is reasonable to hypothesize that smaller models may be able to solve the task. However, exactly how few layers and attention heads per layer are necessary, for each architecture, are not a priori known. By sweeping across these hyperparameters, we find that many smaller models are performant, in some cases outperforming the larger models. And secondly, these sweeps enable us to identify differences between BERT and ALBERT models. These include:
> > 1) BERT models are able to perform well even with fewer layers than the number of elements in the chain (2 layers < 4 elements in the sequence), if there are a sufficient number of attention heads per layer. This again points to the idea that BERT is able to learn shortcut solutions, which ALBERT models, trained with the same architecture, are not able to learn as well.
> > 2) ALBERT consistently outperforms BERT on generalization, as long as there are at least as many layers as there are elements in the sequence (6 layers). Thus, the difference between the two architectures which was found in Fig. 1 is not specific to the large model.
> > 3) ALBERT consistently has greater forward transfer than BERT.
> > 4) Both models consistently do not maintain their performance on the previous experience, when trained on a new experience.
> >
> > We are uncertain by what the reviewer means by the reduced model being suboptimal. For both BERT and ALBERT, the 6 layers - 1 attention head per layer architecture achieves near perfect task accuracy and high forward transfer (Fig. 3A, C, E, G). While there are models with greater generalization accuracy, this architecture is consistent with the general trend of better generalization for ALBERT than BERT (Fig. 3B, F).
> >
> > **"The paper refers to a "generative replay strategy" but there is no generative model being used. The paper states "To demonstrate the potential of generative replay, we train the minimal models in a setting where past experiences are stitched together with new experiences (“incremental flip-flop – Fig. 6A, middle). This serves as an approximate upper bound to the potential success a generative replay strategy might have." What is actually being done is the usage of a simple replay buffer that has been extensively used in the continual learning literature. The results do not go beyond the finding that a replay buffer works in this setting."**
> >
> > We appreciate the reviewer highlighting the confusion surrounding the generative replay strategy. We agree this was not clear and overstated in our original submission. To improve this, we have generally reduced our framing of these results as a “generative replay strategy” in the Abstract, Introduction, Sec. 4.5, and the Discussion. In addition, we have tried to provide more detail on how this approach can be thought of as an upper-bound to generative replay strategies, and what challenges real generative replay strategies might face in the Discussion.

---

### Review · Reviewer_Fw1t · 2026-03-17

**Summary Of Contributions:**

This paper studies the compositional reasoning capabilities of BERT and ALBERT architectures in the continual learning context. It uses the LEGO framework introduced by Zhang et al., and extends it so that the models observe three flip-flop tasks throughout the training. Using this benchmark, the authors compare BERT and ALBERT architectures with different numbers of layers and attention heads. Subsequently, the authors study how using a replay buffer affects the performance of the models. The authors report that ALBERT finds more general solutions (implementing a For loop over elements in the sequence), while BERT finds less generalizable shortcuts. Finally, they compare the model performance under different replay conditions.

Strengths:
- Analyzing the capabilities of compositional reasoning of transformers is an important area of research, as it could help explain the impressive generalization abilities of LLMs.
- The synthetic setup is an elegant and practical way to perform this analysis.
- The paper is well-written and easy to understand.
- The related work, as far as I could tell, has no important missing references.

Weaknesses:
- The paper focuses on BERT and ALBERT architectures which are not used extensively nowadays -- getting a better glimpse into the workings of autoregressive transformer could be significantly more impactful.
- There is a significant overlap between this paper and the original Zhang et al LEGO paper. For example, while the authors examine the impact of the number of layers and attention heads on the results, Zhang et al. also study the impact of the parameter count. The continual setup is indeed fully novel.
- The code is not shared. I wanted to check a couple of details about experimental setting and wasn't able to.
- I feel like certain statements are too strong:
    - While the authors claim that ALBERT exhibits strong forward transfer as opposed to BERT, looking at the plot it seems like the improvement might still be within confidence intervals (though it's difficult to say for sure optically).
    - Also, in order to measure the forward transfer properly, one should not only compare the performance on Exp 2 vs. the performance on Exp 1, but rather compare the performance on Exp 2 of a randomly initialized model and a model pre-trained on Exp 1. That way, one could disentangle the impact of pre-training from properties of Exp 2. In general, I would find it beneficial to see runs on different permutations of {Exp 1, Exp 2, Exp 3} -- do the patterns observed here hold for all orderings?
    - Additionally, authors try to draw conclusions from a single seed: "one of the four seeds trained was not able to learn a4 for the new experiences, demonstrating that such a BERT model had found a suboptimal solution when trained on flip-flop experience 1 that subsequent training is not able to overcome". I don't feel like this is strong enough evidence. If not being able to learn $a_4$ is an event that happens in 25% of cases, there's a chance it also applies to ALBERT but we didn't see it due to a low number of seeds.
    - The assumption that generative replay will allow for compositional data generation is not realistic in my opinion. In the current setup one episode contains data from one experiment (i.e., one flip-flop task), so stitching them together is a form of out-of-distribution generalization that requires additional assumptions.


There are some points that left me confused:
- If the for loop solution, that allows us to solve all the tasks, persists in ALBERT, why do we forget? One hypothesis is we're just missing input-output mapping for the for loop since, as far as I understand from the paper, BERT uses different symbols for each task and the "old" output symbols get forgotten. What would happen in Task-Incremental-Learning-like setup, i.e., separate heads for each task?
- "Average attention" in Fig 4C isn't defined in the caption or the text
- In Figure 5, which values exactly are used to compute the the cosine similarity?
- In Figure 6 it seems like both models have slightly higher performance on $a_6$ than $a_5$. Shouldn't it be that extrapolating further leads to worse results?

**Audience:**

Yes

**Audience Explanation:**

As mentioned in my initial review, I consider compositional research in transformers an important research direction. I feel like the audience would find this paper interesting, even though it only investigates the instantiations of transformer that are currently less popular, i.e., BERT-like architectures.

**Claims And Evidence:**

No

**Claims Explanation:**

While overall the paper does not overstate its main results, I feel like certain statements are too strong, see the corresponding weaknesses section above.

**Requested Changes:**

See the weaknesses section for details, but in general:
- [High importance] Revise the statements or make the experiments stronger on the "certain statements are too strong" point from the weaknesses section
- [Medium importance] I would appreciate a further explanation of the details mentioned in the weaknesses section (what is "average attention" and so on).
- [Minor] Share the code

If the first two points are addressed properly, I would feel comfortable with recommending acceptance of the paper

---

> ### Author Response · Authors · 2026-03-29
> **Rebuttal 1**
>
> We thank the reviewer for their thoughtful and constructive feedback. Below, we address each of their points and reference places in the revised manuscript where changes have been made, which we believe has made our work stronger.
>
> **"The paper focuses on BERT and ALBERT architectures which are not used extensively nowadays -- getting a better glimpse into the workings of autoregressive transformer could be significantly more impactful."**
>
> We agree that our choice in analyzing BERT and ALBERT models is restrictive, particularly given today’s use of autoregressive models. However, to be able to train such autoregressive models on the LEGO task (and, by extension, the continual LEGO task), it would be necessary to modify the training procedure and the way in which the task was structured. We believe this is an important direction to pursue, but would complicate direct comparison of such models with ALBERT and BERT. Therefore, we have decided to stick with just considering BERT and ALBERT. We have added discussion in our revised manuscript of this important limitation in our Abstract, Introduction, and Discussion sections. Additionally, we emphasize that adapting continual LEGO to autoregressive models is an interesting future direction.
>
> **"There is a significant overlap between this paper and the original Zhang et al LEGO paper. For example, while the authors examine the impact of the number of layers and attention heads on the results, Zhang et al. also study the impact of the parameter count. The continual setup is indeed fully novel."**
>
> The Zhang et al. paper was very influential in shaping our work, not only through introducing the LEGO task, but also through their results motivating our consideration of continual learning. For this reason, some of the analyses are similar, although in the context of continual learning. The reviewer correctly points out another similarity: our sweeps over architecture space and Zhang et al.’s study on the impact of parameter count. While in both cases the architecture is being modified, we note that there are two important differences. First, Zhang et al. varied the parameter count of BERT only (their Fig. 17), so as to match the number of parameters in ALBERT. This was done to ensure that the difference between ALBERT and BERT generalization performance was not due to ALBERT having fewer parameters than BERT. In contrast, we vary the architecture of both BERT and ALBERT to understand the architectural dependence of each model. And second, while Zhang et al. studied the effect of parameter count, they maintained the same architecture of BERT (12 layers, 12 attention heads per layer) (their Appendix D.1). This does not address the question of whether such an architecture is strictly necessary, nor does it address questions on how the number of layers and number of attention heads shape performance. By sweeping across the architectural space, we address these questions.
>
> **"The code is not shared. I wanted to check a couple of details about experimental setting and wasn't able to."**
>
> We regret that we have been unable to share our code yet due to funding agreements. However, we would be happy to answer any questions regarding the experimental set-up you might have.
>
> **"Also, in order to measure the forward transfer properly, one should not only compare the performance on Exp 2 vs. the performance on Exp 1, but rather compare the performance on Exp 2 of a randomly initialized model and a model pre-trained on Exp 1. That way, one could disentangle the impact of pre-training from properties of Exp 2. In general, I would find it beneficial to see runs on different permutations of {Exp 1, Exp 2, Exp 3} -- do the patterns observed here hold for all orderings?"**
>
> We agree with the reviewer that forward transfer should be quantified by the comparison of a “pre-trained” model on Experience 1 vs. a randomly initialized model. In fact, this is effectively what we do. Because each flip-flop experience is effectively the same, the number of epochs it takes for the performance to reach 90\% accuracy on Experience 1 - from random initialization - is (in expectation) the same as it takes for the performance to reach 90% accuracy on Experience 2 - from random initialization. Therefore, what is being computed in the equation above is equivalent to the number of epochs needed for the network, pretrained on Experience 1, to reach 90% accuracy on Experience 2, vs. the number of epochs needed for the network, from random initialization, to reach 90% accuracy on Experience 2.  We have added a note of this in Appendix C of the revised manuscript.

---

> > ### Author Response · Authors · 2026-03-29
> > **Rebuttal 2**
> >
> > **"Additionally, authors try to draw conclusions from a single seed: "one of the four seeds trained was not able to learn a4 for the new experiences, demonstrating that such a BERT model had found a suboptimal solution when trained on flip-flop experience 1 that subsequent training is not able to overcome". I don't feel like this is strong enough evidence. If not being able to learn is an event that happens in 25% of cases, there's a chance it also applies to ALBERT but we didn't see it due to a low number of seeds."**
> >
> > This is a good point and we agree that we overstated. We have weakened our conclusions and removed some of the text on this in the revised manuscript .
> >
> > **"The assumption that generative replay will allow for compositional data generation is not realistic in my opinion. In the current setup one episode contains data from one experiment (i.e., one flip-flop task), so stitching them together is a form of out-of-distribution generalization that requires additional assumptions."**
> >
> > We appreciate the reviewer pointing out this confusion. We agree that each episode consists of data from a single experience. However, we note that in this case, the experiences are connected by a common element. In particular, Experience 1 comprises flip-flopping between elements 1 and 2, and Experience 2 comprises flip-flopping between elements 1 and 3 (Fig. 6A). If we have only observed Experience 1, then we would model the generative process that samples the chains as consisting only of going between 1 and 2. However, if we have seen both Experiences 1 and 2, then we can model the generative process as being equally likely to go from 1 to 2 as from 1 to 3. From that, new sequences could be generated. But importantly, we would need a way to identify what the new experience “rules” are, as quickly as possible in training. We have added discussion of this challenge for true generative replay approaches to the Discussion of the revised manuscript, and have generally reduced our framing these results as a “generative replay strategy” in the Abstract, Introduction, Sec. 4.5, and the Discussion.
> >
> > **"If the for loop solution, that allows us to solve all the tasks, persists in ALBERT, why do we forget? One hypothesis is we're just missing input-output mapping for the for loop since, as far as I understand from the paper, BERT uses different symbols for each task and the "old" output symbols get forgotten. What would happen in Task-Incremental-Learning-like setup, i.e., separate heads for each task?"**
> >
> > We agree with the reviewer that the forgetting, even for ALBERT, suggests that there is a missing input-output map, which is driven by the new experience. We believe this is why the replay buffer, even at a low percent, is so effective (Fig. 5). However, that the replay buffer, by itself, is not sufficient to enable generalization when the task requires composing across experiences (e.g., going from 2 -> 1 -> 3 -> 1 -> 4 -> 1 in Fig. 6A) suggests that, in addition to the input-output mapping being disrupted across experiences, the models can develop solutions that do not “connect” different experiences. That is, the model learns a solution where Experience 1 and Experience 2, despite having one of the same elements, are not “viewed” as being related. We believe this is why the Incremental task is so important, as it forces the model to learn to connect multiple experiences by training on chains that are sampled from Experience 1 and 2.
> >
> > **" "Average attention" in Fig 4C isn't defined in the caption or the text
> > We thank the reviewer for pointing out this omission. We have added a definition of "average attention” to the revised manuscript in Fig 4 caption."**
> >
> > **"In Figure 5, which values exactly are used to compute the cosine similarity?"**
> >
> > For calculation of attention cosine similarity, the two-dimensional attention pattern for a model in response to an input sample is first flattened into a vector. Cosine similarity is then calculated between models on these flattened attention vectors. We have added this clarification to the caption of Figure 5.
> >
> > **"In Figure 6 it seems like both models have slightly higher performance on a6 than a5. Shouldn't it be that extrapolating further leads to worse results?"**
> >
> > This is a good observation and one that was made in original LEGO paper (Zhang et al. - their Sec. 4.3). “Surprisingly, we find that the Rand-Init BERT and ALBERT models first learn a “shortcut” solution: they immediately resolve the last variable in the reasoning chain, perhaps by counting the total number of minus signs”. We believe this is likely what's behind the higher accuracy for a_6 than a_5, particularly in the 0% replay setting.

---

### Decision · Action_Editor_g6JM · 2026-05-04

**Recommendation:** Reject

**Audience:**

Yes

**Audience Explanation:**

The reviewers and I agree that another careful study of compositional generalization in transformers in controlled setups would be of interest to TMLR readers.

Although there is substantial overlap with ref. 1, my opinion is that the paper would meet the bar for publishing at TMLR were the paper's analyses significantly strengthened and its methodological issues resolved, especially given its novel focus on continual learning.

**Claims And Evidence:**

No

**Claims Explanation:**

The reviewers raised several claim backing issues that I think are critical, in particular:
- Weak/potentially insignificant forward transfer effects, as pointed out by Reviewer Fw1t.
- Arbitrary hyperparameter selection, as pointed out by Reviewer W8Qe.
- The mechanistic analyses are suggestive but not strong enough to back the paper's claims, as pointed out by Reviewer W8Qe.

The issues should be solved before this paper is ready for publication at TMLR.

**Resubmission Of Major Revision:**

The authors may consider submitting a major revision at a later time.